# A Differentiable Dynamic Model for Musculoskeletal Simulation and Exoskeleton Control

**DOI:** 10.3390/bios12050312

**Published:** 2022-05-09

**Authors:** Chao-Hung Kuo, Jia-Wei Chen, Yi Yang, Yu-Hao Lan, Shao-Wei Lu, Ching-Fu Wang, Yu-Chun Lo, Chien-Lin Lin, Sheng-Huang Lin, Po-Chuan Chen, You-Yin Chen

**Affiliations:** 1Department of Biomedical Engineering, National Yang Ming Chiao Tung University, Taipei 11221, Taiwan; chaohungk@gm.ym.edu.tw (C.-H.K.); jiaweich@nycu.edu.tw (J.-W.C.); ianyang01@nycu.edu.tw (Y.Y.); leolan9534766@nycu.edu.tw (Y.-H.L.); eeg.be09@nycu.edu.tw (S.-W.L.); chingfu.wang@nycu.edu.tw (C.-F.W.); 2School of Medicine, National Yang Ming Chiao Tung University, Taipei 11221, Taiwan; 3Department of Neurological Surgery, Neurological Institute, Taipei Veterans General Hospital, Taipei 11217, Taiwan; 4Department of Neurological Surgery, University of Washington, Seattle, WA 98195-6470, USA; 5Biomedical Engineering Research and Development Center, National Yang Ming Chiao Tung University, Taipei 11221, Taiwan; 6The Ph.D. Program for Neural Regenerative Medicine, College of Medical Science and Technology, Taipei Medical University, Taipei 11031, Taiwan; aricalo@tmu.edu.tw; 7Department of Physical Medicine and Rehabilitation, China Medical University Hospital, Taichung 404332, Taiwan; d5699@mail.cmuh.org.tw; 8School of Chinese Medicine, College of Chinese Medicine, China Medical University, Taichung 406040, Taiwan; 9Department of Neurology, Hualien Tzu Chi Hospital, Buddhist Tzu Chi Medical Foundation, Hualien 97002, Taiwan; 10Department of Neurology, School of Medicine, Tzu Chi University, Hualien 97004, Taiwan; 11School of Electrical and Computer Engineering, Georgia Institute of Technology, Atlanta, GA 30332, USA; pchen353@gatech.edu

**Keywords:** differentiable physics, electromyography (EMG), musculoskeletal model, Hill-type muscle, exoskeleton, motor control, adjoint method, gradient, differential equation

## Abstract

An exoskeleton, a wearable device, was designed based on the user’s physical and cognitive interactions. The control of the exoskeleton uses biomedical signals reflecting the user intention as input, and its algorithm is calculated as an output to make the movement smooth. However, the process of transforming the input of biomedical signals, such as electromyography (EMG), into the output of adjusting the torque and angle of the exoskeleton is limited by a finite time lag and precision of trajectory prediction, which result in a mismatch between the subject and exoskeleton. Here, we propose an EMG-based single-joint exoskeleton system by merging a differentiable continuous system with a dynamic musculoskeletal model. The parameters of each muscle contraction were calculated and applied to the rigid exoskeleton system to predict the precise trajectory. The results revealed accurate torque and angle prediction for the knee exoskeleton and good performance of assistance during movement. Our method outperformed other models regarding the rate of convergence and execution time. In conclusion, a differentiable continuous system merged with a dynamic musculoskeletal model supported the effective and accurate performance of an exoskeleton controlled by EMG signals.

## 1. Introduction

The exoskeleton, a wearable device, was designed according to the physical and cognitive interactions of the users. Exoskeletons have been developed for diverse medical purposes, including strength augmentation [1], rehabilitating neurological impairment [2], and assisting daily activities caused by neuro-musculoskeletal disorders [3]. In each application, the design of the exoskeleton would be based on the understanding of the users’ intention to control the device to achieve precise execution of the intended movement. 

Thus, for patients with stroke or spinal cord injury, decoding motor signals from electromyography (EMG) or neural signals from electrocorticography (ECoG) [4,5] would be crucial for controlling exoskeletons and helpful by assisting these patients in restoring daily functions. The effect of the muscle spindle on muscle tension can be mathematically quantified as a neuromuscular model and combined with control theory to optimize the system for a human limb to operate the device [6,7]. This creates the potential for developing the exoskeleton-assisted rehabilitation approach into a personalized therapy.

For controlling an exoskeleton, several physiological signal sources, including force sensors [8] or biomedical signals, such as EMG [9] and electroencephalography (EEG) [10], were used with excellent performance. In the preceding study, the force sensor systems used over hip-mounted exoskeletons for elders with walking difficulties caused by muscle weakness. The results demonstrated that the sensor system can generate torque in the walking-assistant actuator from the direct measurement of the hip-assistance force and that the gait-assistance exoskeleton system can control the delivered power and torque for the user [8]. 

However, in that study, the coordination and interaction between physical activity and cognitive control was limited for the force sensor system because the predetermined trajectory could not synchronize the user’s intension to actively coordinate the muscle movements simultaneously. The force sensor system could only be activated by the predetermined trajectory of the exoskeleton when the actual output was lower than the threshold or deviated from the predetermined trajectory. The unsynchronized interaction, as well as a large time delay, between physical movements and the force sensors increased the risk of tripping [11]. 

For EEG to be used as the input to control the exoskeleton, direct brain signal recording has been measured on the scalp. In previous studies, the exoskeleton could be controlled via the brain-computer interface by using EEG signals [12,13]. However, because of brain pathology, including stroke, Parkinson’s disease, or brain injury, EEG signals varied across subjects and their use had to be highly customized and needed a long learning period, rendering the approach inefficient for real-time applications [14]. This necessitated the design of a control algorithm that can detect the user’s intention fast and does not rely on a predetermined trajectory.

To overcome the limitations posed by the time delay between the user’s intentions and the force sensor system, it is crucial to directly decode biomedical signals recorded from the nervous system or the musculoskeletal system. EMG signals are relatively easy to obtain and contain the information about body movement. Other investigators have used machine learning or neural networks to learn and predict the joint angles [15] and torques [16,17,18] of the human body during movement. 

In one example, the artificial neural network technique was applied to EMG signals to predict the joint torque estimation model, and the trained model was used in the arm rehabilitation device aiding limb-paralyzed patients. Although the methods of machine learning or neural networks had performed well on decoding, their “black-box” transformation of the biomedical signals was lacking biomechanical explanation, which remains a source of controversy in these applications.

The Hill-type muscle model [19], built of contractile, series elastic, and parallel elastic components, offers a clear biomechanical explanation of the mechanical responses representing muscle behaviors [20,21]. An application of the Hill-type muscle model was shown to be capable of accurately simulating gait in a three-dimensional (3D) model, according to the EMG signals recorded from healthy populations [22]. 

For a more advanced application to not only healthy subjects but also paralyzed patients, a Hill-type muscle model, complete with output activation and torque-angles, was used to determine the muscle parameters for biomechanical computer simulations. This method was able to provide adaptive control and generate muscle responses in real-time. Thus, the Hill-type muscle model implements a biomechanical mechanism for natural and reliable interaction between assistance and estimation [23,24], and could also be considered in the design of exoskeleton control [24,25]. 

Although the control of neuromuscular stimulation and the exoskeleton can be done cooperatively, it requires an accurate and robust system to identify the parameters. To improve the accuracy of the identified parameters, an iterative method of the sequential least-squares quadratic programming (SLSQP) [26,27,28] with approximate gradients and second-order derivatives was used to optimize the parameters of the musculoskeletal model. However, using gradient approximation in the musculoskeletal model for parameter tuning and muscle force estimation required more iterations, and therefore more computational time. 

To make it practicable, the musculoskeletal model had to be simplified [29]. Due to the challenges of analyzing complex models, gradient-free methods such as linear optimization, genetic algorithms [30,31], and the Nelder–Mead (NM) algorithm [11,23,25] were often used to optimize parameters in the musculoskeletal models. However, the adjusted parameters obtained with gradient-free methods can easily diverge, making the initialization of parameters challenging.

To tackle the challenges described above, we introduced the method of differentiable systems [32] applied to spring-damper models in 2D and 3D, which also was used for adjoint optimization on 3D rigid body engines [33], and the integration of articulated body dynamics into deep-learning frameworks and gradient-based optimization of neural networks that operated on articulated bodies [34]. Our proposed model was tested with healthy people for control of the knee exoskeleton in flexion–extension movements, which merged a differentiable continuous system with a dynamic musculoskeletal model to support an effective and accurate performance for an exoskeleton controlled by EMG signals. 

The main biological and engineering contributions of this study are as follows. First, we propose a differentiable continuous system control that achieves assist-as-needed control by integrating the dynamics of musculoskeletal models and analyzing the interaction between the user and the exoskeleton. Second, in this continuous system, we implement an analytical gradient and updating mechanism for the muscular excitation parameters. As a result, and in contrast to previous gradient approximation methods and gradient-free methods, our proposed model is capable of providing the robust and accurate gradients needed to obtain the parameters in the musculoskeletal model.

## 2. Materials and Methods

### 2.1. Myoelectric Processor

Subjects used a wearable and wireless EMG amplifier (Artise Biomedical Co., Ltd., Taiwan) equipped with eight bipolar signal channels that were connected with medical grade button snap cables, common in clinical practice, to pairs of surface electrodes (Red Dot™, 3M, Maplewood, MN, USA) located on the targeted muscle tissues. Body ground was applied on the leg to increase the signal-to-noise ratio. 

The acquired EMG signals were amplified by a 12× programmable gain, sampled at 1000 Hz, digitized at 24-bit with delta-sigma ADC (Texas Instruments, Dallas, TX, USA), and transmitted to a customized radio-frequency receiver dongle connected to the computer as a COM port input. The baud rate was configured at 921,600 bits/s to enable high-speed transmission. Sampled data was labeled by time stamp and channel counter, passed through an error-control mechanism [35] to achieve highly synchronized and simultaneous recording. The raw EMG signal filtered in the DC to 262 Hz frequency pass-band was recorded and streamed to a real-time processing program for exoskeleton feedback. 

### 2.2. Experimental Setup

In this study, the knee exoskeleton (KneeBO^TM^, FREE Bionics Taiwan Inc., Hsinchu, Taiwan) was applied to a single knee joint in a human-in-the-loop environment as shown in Figure 1. The experimental environment combined the EMG signals, the angle of torque from the exoskeleton, and the angle for knee movement of the subject into a single computational platform.

The purpose of the exoskeleton was to support flexion and extension movements involving the knee joint. The subject was in the sitting position with the hip and knee flexed naturally as shown in Figure 2A. There are eight anatomically separate muscle–tendon units supporting the knee joint, including the rectus femoris (RF); the vastus muscle (VM) group, including the vastus lateralis, vastus medialis, and vastus intermedius; the semitendinosus; the semimembranosus; and the biceps femoris (BF), including both short and long heads. 

While the semitendinosus was assumed to have the same activation as the semimembranosus [36], the semitendinosus muscle group (ST) included the semitendinosus and the semimembranosus. In this study, four pairs of surface EMG electrodes were positioned at skin locations above the muscles of RF, VM, BF, and ST as shown in Figure 2B,C, which were used to measure the EMG activity during flexion–extension movements of knee. The study was reviewed and approved by the Taipei Veterans General Hospital Institutional Review Board (IRB2020-10-001). All recoded EMG data are made available at the repository (https://doi.org/10.5281/zenodo.6516777), assessed on 29 April 2022.

During data acquisition, signals from the exoskeleton and the EMG electrodes were recorded simultaneously. First, the subject was asked to release the force. The knee in full extension (0 degrees) and flexion (100 degrees) was completely supported by the exoskeleton. Furthermore, the subject could move the knee freely without support from the exoskeleton. 

After analyzing the parameters of the exoskeleton and EMG signals, we evaluated the performance (validation phase) while the subject was moving the knee with simultaneous exoskeleton support. The schematics of the experiment is shown in Figure 3, and the details of the methods are described in the sections below. A summary table to identify the corresponding physiological interpretations of the inertial parameters for the differentiable musculoskeletal model is presented in Table 1.

### 2.3. Preprocessing

To eliminate movement artifacts, the EMG data acquired from the four muscles were processed in the following three steps. First, the raw EMG data were filtered by a fourth-order Butterworth band-pass (20–450 Hz) filter. Second, full-wave rectification and an 8-Hz low-pass second-order Butterworth filter were applied to obtain the EMG signal envelope. Third, the filtered EMG signal was normalized with respect to the muscle’s maximum voluntary contraction level. 

After the above three steps the smoothed normalized EMG signal e(t), with an amplitude range in [0, 1], was used to determine the neural activation u(t) and muscle activation a(t). Since muscle activation takes time to generate force, there is a time lag between the two. The delayed transformation of EMG signals e(t) to neural activation u(t) was formalized by a damped linear second-order differential equation expressed in discrete form as a recursive filter (Equation (1)) [36,37],
(1)$$u\left(t\right)=\alpha e\left(t-d\right)-\beta_{1}u\left(t-1\right)-\beta_{2}u\left(t-2\right),$$
where α represents the muscle gain coefficient; $\beta_{1}$ and $\beta_{1}$ are the muscle recurrence coefficients; and d is the electromechanical delay. At the motor unit level, the increase in muscle force was associated with an exponential increase in the firing rate. Equation (2) formalizes this nonlinear relationship between neural activation *u*(*t*) and muscle activation *a(t)*, where A represents the nonlinear shape factor with the −3<A<0 constraint.
(2)$$a\left(t\right)=\frac{exp^{Au\left(t\right)}-1}{exp^{A}-1}$$

Since muscle torque cannot be measured invasively, the torque obtained from the motion equation was used as the substitute when adjusting the muscular parameters. This required the three inertia parameters of the motion equation (the rotation inertia, J; the damping, B; and the stiffness, K, of the knee joint at full extension of the lower-limb) to be identified in advance. The motion equation, illustrated in Figure 2A, is defined in Equation (5) below. The identification process was conducted without human-exerted muscle torque, i.e., $\tau_{h}\left(t_{i}\right)=0$, while the subject’s lower-limb was moved by the exoskeleton.

First, K was determined in static conditions as the exoskeleton moved from angular position $\theta =120^{\circ }$ to $0^{\circ }$ in $15^{\circ }$ steps. In static conditions, the velocity $\dot{\theta }$ and acceleration $\ddot{\theta }$ of the exoskeleton equals 0 by definition: $\dot{\theta }\left(t_{i}\right)=\ddot{\theta \left(t_{i}\right)}=0$. With these values substituted into the motion equation (Equation (5)), the identification of K can be expressed as Equation (3): (3)$$\begin{array}{ll} \underset{K}{argmin} & \;\frac{1}{2}\sum_{i=1}^{N}∥y\left(t_{i}\right)-K\left(\theta \left(t_{i}\right)-\theta_{r}\right){∥}^{2} \end{array},$$
where $y\left(t_{i}\right)=\tau_{e}\left(t_{i}\right)-\tau_{g}\left(t_{i}\right)\sin\left(\theta \left(t_{i}\right)-\theta_{r}\right)$ is the conversion of the gravitational torque, $\tau_{g}\left(t_{i}\right)=m\cdot g\cdot r_{cm}$, and the exoskeletal torque, $\tau_{e}\left(t_{i}\right)$, from the electrical current sensor representation on the exoskeleton. Here, $m,g,r_{cm},\;\mathrm{and}\;\theta_{r}$ are the mass, gravity, moment arm, and rest position, respectively, while N is the number of acquired samples, and $t_{i}$ is the time of the i-th acquired sample. 

Next, J and B were determined in dynamic conditions, while the exoskeleton was moving from $\theta =90^{\circ }$ to $0^{\circ }$ at a variable frequency between 0.01 and 0.5 Hz. The identification of these two parameters, similarly to that of K, is formalized in Equation (4),
(4)$$\begin{array}{ll} \underset{J,B}{argmin} & \;\frac{1}{2}\sum_{i=1}^{N}∥y\left(t_{i}\right)-\left(J\ddot{\theta }\left(t_{i}\right)+B\dot{\theta }\left(t_{i}\right)\right){∥}^{2} \end{array},$$
where $y\left(t_{i}\right)=\tau_{e}\left(t_{i}\right)-\tau_{g}\left(t_{i}\right)\sin\left(\theta \left(t_{i}\right)-\theta_{r}\right)-K\left(\theta \left(t_{i}\right)-\theta_{r}\right)$. The optimization problems in Equations (3) and (4) were solved by the Levenberg–Marquardt algorithm. Since the motor encoder on the exoskeleton can only measure the angular position and velocity with noise, we used the parameterized Fourier series to calculate the smooth (“noise-free”) angular velocity and acceleration [38].

### 2.4. Wearable Exoskeleton Modeling

Since the motion of the knee exoskeleton is similar to the pendulum, the model dynamics can be derived using the same principles of mechanics. Using Lagrangian mechanics [39,40], the motion equation of the knee exoskeleton is derived as Equation (5),
(5)$$\tau \left(t\right)=J\ddot{\theta }\left(t\right)+B\dot{\theta }\left(t\right)+\frac{\partial E_{p}}{\partial \theta }=\tau_{e}\left(t\right)+\tau_{h}\left(t\right),$$
where the potential energy, $E_{p}=\frac{1}{2}K{\left(\theta \left(t\right)-\theta_{r}\right)}^{2}-\tau_{g}\cos\left(\theta \left(t\right)-\theta_{r}\right)$, corresponds to the resting position $\theta_{r}=\frac{\pi }{2}$, and the inertial parameters J,B and K were as described in the preprocessing. The exoskeleton torque, $\tau_{e}\left(t\right)$, was converted from the motor current.

The total muscle torque $\tau_{h}\left(t\right)$ is estimated by Equation (6), where the superscript j denotes the j-th muscle.
(6)$$\tau_{h}\left(t\right)=\overset{8}{\underset{j=1}{\sum }}F_{mt}^{\left(j\right)}\left(a^{\left(j\right)}\left(t\right),l^{\left(j\right)}\left(t\right)\right),$$

In the Hill-type muscle model, the $F_{mt}$ force generated by the muscle–tendon contraction was mainly determined by the circuit composed of a contractile element (CE) and two non-linear spring elements, one in parallel element (PE) and another in series element (SE) [19,41] as formalized in Equation (7):(7)$$\begin{array}{ll} & F_{mt}\left(a\left(t\right),l\left(t\right)\right)=F_{se}\left(t\right)\\ & =F_{m}\left(t\right)\cos\left(\alpha \right)\\ & =\left[F_{ce}\left(t\right)+F_{pe}\left(t\right)\right]\cos\left(\alpha \right)\\ & =\left[f_{l}\left(l\left(t\right)\right)f_{v}\left(v\left(t\right)\right)a\left(t\right)+f_{pe}\left(l\left(t\right)\right)\right]F_{\max}\cos\left(\alpha \right) \end{array}$$

For simplicity, we only present the equation derived for one muscle that lets us omit the superscript j. In this model, α represents the pennation angle; l(t) represents the normalized muscle length, calculated as the current muscle length divided by the length while the muscle contraction is the maximum ($l\left(t\right)=\frac{L\left(t\right)}{L^{⋆}}$); and v(t) represent the normalized velocity of muscle contraction ($v\left(t\right)=\frac{\dot{L}\left(t\right)}{L^{⋆}}$), i.e., the change of the muscle length.

In this study, a Hill-type muscle model can be simulated by the active (contractile) and passive (noncontractile) muscle components [42,43]. The relationship between contraction force and muscle length (l(t)) is formalized by the Gaussian function (Equation (8)) for active muscle contraction [44] and an exponential function (Equation (9)) for passive muscle contraction [45], where γ, k, and ε were shape parameters of the functions.
(8)$$f_{l}\left(l\left(t\right)\right)=exp\left(-\frac{{\left(l\left(t\right)-1\right)}^{2}}{\gamma }\right)\;$$
(9)$$f_{pe}\left(l\left(t\right)\right)=\frac{exp^{\frac{k\left(l\left(t\right)-1\right)}{\varepsilon }}-1}{exp^{k}-1},\;$$

The force–strain relationship of a tendon is determined by an exponential function within an initial nonlinear toe region followed by a linear function outside (Equation (10)),
(10)$$f_{se}\left(l_{t}\left(t\right)\right)=\{\begin{array}{ll} \frac{f_{toe}}{exp^{k-1}}\left(exp^{\frac{k_{toe}\varepsilon }{\varepsilon_{toe}}}-1\right), & \varepsilon \leq \varepsilon_{toe}\\ k_{lin}\left(\varepsilon -\varepsilon_{toe}\right)+f_{toe}, & \varepsilon >\varepsilon_{toe} \end{array}$$
where $l_{t}\left(t\right)$ represents tendon strain; $\varepsilon_{toe}$ is the threshold tendon strain above which the tendon exhibits linear behavior; $k_{toe}$ is the exponential shape factor; $k_{lin}$ is the linear shape factor, and ε is the strain normalized by $l_{slack}$, the slack length, and calculated as $\frac{\varepsilon =l_{t}\left(t\right)}{l_{slack}}$.

The formal relationship between the v(t) muscle contraction velocity and the $F_{t}\left(t\right)$ force arising from muscle–tendon contraction can be derived from by Equations (7)–(10) as stated in Equation (11):(11)$$f_{v}\left(v\left(t\right)\right)=\frac{F_{se}\left(t\right)-f_{pe}\left(l\left(t\right)\right)}{a\left(t\right)\cdot f_{l}\left(l\left(t\right)\right)}\;$$

The v(t) muscle contraction velocity is determined from the inverse transform of Equation (11) and formalized, after some modification [41], in Equation (12): (12)$$\begin{array}{ll} & v\left(l\left(t\right),f_{ce}\left(t\right),a\left(t\right)\right)=\left(0.25+0.75a\right)V_{\max}\frac{f_{ce}\left(t\right)-a\left(t\right)f_{l}\left(l\left(t\right)\right)}{b} \end{array}$$
where a and b are shape parameters.

The force of active muscle contraction is derived, in turn, from the inverse transform of from Equation (12), i.e., by Equation (13):(13)$$f_{ce}\left(t\right)=v^{-1}\left(f_{l}\left(l\left(t\right)\right),a\left(t\right)\right)$$

To this end, the force of muscle–tendon contraction required by Equation (7) is demonstrated as $F_{mt}\left(a\left(t\right),l\left(t\right)\right)=\left[f_{ce}\left(t\right)+f_{pe}\left(l\left(t\right)\right)\right]F_{\max}\cos\left(\alpha \right)$ and is adopted for each muscle in Equation (6).

However, each user had different muscle properties; therefore, each muscle in the model needs to be calibrated for the parameters. The Hill’s muscle model requires three parameters to scale generic curves for active and passive force generation: the optimal fiber length $L^{⋆}$, maximum isometric force $F_{\max}$, and tendon slack length $l_{slack}$ [46]. The *adjoint* method [47], which we describe next, is used to determine these parameters.

### 2.5. Differentiable Musculoskeletal Parameters Estimation

Muscle torque cannot be measured directly in an invasive manner. To estimate muscle torque, it was necessary to use statistical inference consistent with physiological and physical explanations. Under the condition that the knee joint of the subject is in the same position as the exoskeleton joint, rearranging Equation (5) yields the functional relationship between muscle torque and the angular acceleration of the joint generated by muscle torque as stated in Equation (14):(14)$$h\left(\tau_{h}\left(t\right)\right)=J^{-1}\left(\tau_{h}\left(t\right)-B\dot{\theta }\left(t\right)-\frac{\partial E_{p}}{\partial \theta }\right)$$

The angular position of the trajectory can be computed using this equation from the inputs of muscle torque $\tau_{h}\left(t\right)$00 as part of the forward dynamics model. The muscle skeleton model is formalized as a system of differential equations (Equation (15)):(15)$$\{\begin{array}{c} \frac{dl\left(t\right)}{dt}=v\left(l\left(t\right),f_{ce}\left(t\right),a\left(t\right)\right)\\ \frac{d\dot{\theta }\left(t\right)}{dt}=h\left(\tau_{h}\left(t\right)\right)\\ \frac{d\theta \left(t\right)}{dt}=\dot{\theta }\left(t\right) \end{array}$$

In this system of differential equations, the first component, the velocity of muscle contraction, is calculated from Equation (12). The second component, angular acceleration, requires the muscle length, integrated from first component, to compute the muscle torque $\tau_{h}\left(t\right)$ from Equation (6). The third component, angular velocity, is integrated from the second component. This set of differential equations can be solved using the Runge–Kutta–Fehlberg method for predicting the system state. During motion, the assistive torque, $\tau_{e}\left(t\right)$, is regulated by the motor’s proportional–integral–derivative (PID) controller (Equation (16)). In control theory, the PID controller is the mature method, and its formal analysis, such as via the Laplace transformation, can refer to Equation (16) [48].
(16)$$\tau_{e}(t)=K_{p}\hat{e}(t)+K_{i}\int_{0}^{t}\hat{e}(\mu )d\mu +K_{d}\frac{d\hat{e}\left(t\right)}{dt}$$

In the PID controller, $K_{p}$ represents the proportional gain; $K_{i}$ is the integral gain; and $K_{d}$ is the derivative gain; these three are tuning parameters. The tracking error, $\hat{e}\left(t\right)$, also known as the control feedback, is the difference between the target position and the position predicted by integrating Equation (15), where the target position $\theta^{*}\left(t\right)$ is manually specified in advance.

To estimate the unmeasurable muscle torque, $\tau_{h}\left(t\right)$, from the measurable angular velocity of trajectory, $\dot{\theta }\left(t\right)$, the model was set up by maximum-a-posteriori (MAP) estimation, and maximized posterior probability (Equation (17)):(17)$$\begin{array}{ll} \hat{\tau_{h}} & =\underset{\tau_{h}}{\mathrm{argmax}}\;p\left(\tau_{h}|\dot{\theta }\right)\\ & =\underset{\tau_{h}}{\mathrm{argmin}}-\log\;p\left(\dot{\theta }|\tau_{h}\right)-\log\;p\left(\tau_{h}\right), \end{array}$$
where $\tau_{h}$ and $\dot{\theta }$ represent the $\tau_{h}\left(t\right)$ and $\dot{\theta }\left(t\right)$ time series, respectively.

In Equation (17), the prior probability $-\log p\left(\tau_{h}\right)$ was set to 0, signifying that we did not make any assumptions about muscle torque. The conditional probability $-\log p\left(\dot{\theta }|\tau_{h}\right)$ requires computing the forward dynamics model, which implies calculating the optimally angular velocity $\dot{\theta }\left(t\right)$ from the unmeasurable muscle torque $\tau_{h}\left(t\right)$. To overcome this obstacle, the conditional probability was recast as the integrals ℓ(⋅) (Equation (18)):(18)$$\ell \left(\dot{z},\dot{\theta }\right)=\int_{0}^{t}∥\dot{z}\left(\mu ,\tau_{h}\right)-\dot{\theta }\left(\mu \right){∥}^{2}d\mu ,\;$$
and $\dot{z}\left(\cdot \right)$ (Equation (19)), which was used to explain the angular velocity of the muscle skeleton by integrating Equation (15):(19)$$\dot{z}\left(t,\tau_{h}\right)=h\left(\tau_{h}\left(t_{0}\right)\right)+\int_{0}^{t}h\left(\tau_{h}\left(\mu \right)\right)d\mu \;$$

Here, $\tau_{h}$ was modeled by Hill’s muscle (Equation (6)), and was adjusted based on each subject’s muscle parameters [46]. The muscle parameters for adjusting the muscle torque included optimal fiber length, maximum isometric force, and tendon slack length, respectively ($\omega =\left\{\;L^{⋆},F_{\max},l_{slack}\;\right\}$). With Equations (18) and (19), the MAP estimation of Equation (17) becomes equivalent to solving the constrained optimization problem, formalized by Equation (20) below:(20)$$\begin{array}{ll} \underset{\omega }{argmin}\; & \frac{1}{N}\sum_{i=1}^{N}\ell \left({\dot{z}}_{i},{\dot{\theta }}_{i}\right)\\ \;s,t.\; & \frac{d\dot{z}\left(t_{i}\right)}{dt}=h\left(\tau_{h}\left(t_{i},\omega \right)\right),\;h\left(\tau_{h}\left(t_{0}\right)\right)=\ddot{\theta }\left(t_{0}\right)\\ & {\dot{z}}_{i}=\dot{z}\left(t_{i},\tau_{h}\right),\;i=1,2,.N, \end{array}$$
where, assuming that N samples were acquired, ${\dot{z}}_{i}$ represents the angular velocity of the i-th acquired sample, and $t_{i}$, the time of the i-th acquired sample.

In order to solve Equation (20) using an iterative algorithm, such as a stochastic gradient descent or adaptive moment estimation (Adam) [49], it was necessary to derive the gradient of the muscle parameters ω. We used the Lagrangian Multiplier method to obtain the Lagrangian ℒ (Equation (21)) from Equation (20),
(21)$$ℒ=\ell \left(\dot{z},\dot{\theta }\right)+\int_{0}^{t}\lambda {\left(\mu \right)}^{⊤}\left[\frac{d\dot{z}\left(\mu \right)}{dt}-h\left(\tau_{h}\left(\mu ,\omega \right)\right)\right]d\mu \;$$
and then used the Karush–Kuhn–Tucker conditions to derive the gradient. The detailed derivation can be found in the *adjoint* method [47]. In brief, the analytical gradient can be calculated by the algorithm consisting of these four steps: (1) Solve $\dot{z}\left(t,\tau_{h}\right)$ from time 0 to t  (Equation (19)). (2) Determine λ(t) (Equation (22)). (3) Solve λ(t) from time t to 0 (Equation (23)). (4) Calculate the Lagrangian gradient in the muscle parameters (Equation (24)). For simplicity, only the equations resulting for N=1 are presented, and the subscript i is omitted:(22)$$\frac{\partial \ell }{\partial \dot{z}\left(t\right)}+\lambda \left(t\right)=0$$
(23)$$\frac{d\lambda \left(t\right)}{dt}+{\left(\frac{\partial h\left(\tau_{h}\left(t,\omega \right)\right)}{\partial \dot{z}\left(t\right)}\right)}^{⊤}\lambda \left(t\right)=0$$
(24)$$\frac{\partial ℒ}{\partial \omega }=\int_{t}^{0}\lambda {\left(\mu \right)}^{⊤}\frac{\partial h\left(\tau_{h}\left(\mu ,\omega \right)\right)}{\partial \omega }d\mu$$

The performance of our proposed model was evaluated in two different sets of tests. In the first set of tests, conducted under muscle contraction, the torques and angles predicted in each muscle were compared with real data. In the second set of tests, the efficiency of our proposed method was evaluated by the rate of convergence and execution time and compared with those of other models on an equal basis.

## 3. Results

### 3.1. Parameters Estimated in the Motion Equation

In exoskeleton control, the joint torque generated by the muscle force was estimated to predict the movement trajectory. To obtain an estimate of the muscle-generated torque in a non-invasive manner, we identified the parameters of the motion equation (Equation (5)) in advance from the recorded angular position and velocity of the trajectory. The five identified parameters in the motion equation are summarized in Table 2. Three of the parameters (the inertia J, damping B, and stiffness K) were calculated as described in the preprocessing. The remaining two, the mass m and the center of mass $r_{cm}$ of the exoskeleton, were measured manually.

Table 3 summarized the estimates for the three muscle parameters used in the Hill’s model (Equation (7)), including optimal fiber length $L^{⋆}$, tendon slack length $l_{slack}$, and maximum isometric force $F_{\max}$. The first column lists the anatomical grouping of the eight muscles from which the EMG signals were obtained via four electrodes. The values of the muscle parameters that we identified were similar to those obtained by physiological measurements in previous studies [46].

The process used for the optimization of the muscle parameters is illustrated for predicted position in Figure 4A. Initially, the predicted position obtained using the unadjusted muscle parameters (solid blue line) was a poor approximation of the reference trajectory generated by the subject (dashed line). As the number of iterations increased, the estimated muscle parameters were better adapted to the subject’s generated trajectory. A similar iterative improvement was observed in the predicted torque as shown in Figure 4B).

### 3.2. Prediction Results

In the model simulation, the time interval of the prediction ranged from 2 to 25 s, the EMG signals were served as inputs to the model, and the model outputs were the predicted trajectory and joint torque. The solid line in Figure 5A represents the predicted trajectory obtained by numerical integration of the third term of Equation (15). The dashed line represents the trajectory measured from the subject used as the reference trajectory. The goal of our optimization was to adjust the muscle parameters so that the predicted trajectory was an increasingly good approximation to the reference trajectory. 

The predicated joint torque (Figure 5B) was calculated from Equation (6) by using the muscle activation and predicted muscle length. The predicted torque was becoming increasingly similar to the reference torque, mirroring what was seen for the predicted trajectory. Figure 5C,E show, for the four muscle groups defined in Table 3, the average muscle length predicted by numerical integration (first term in Equation (15)). Similarly, Figure 5D,F shows the muscle activation, a(t), calculated from the EMG signals using Equation (2).

### 3.3. Performance Analysis: Convergence and Execution Time

We compared the performance of the analytical gradient method and three other gradient update methods. Two gradient-free methods, Nelder–Mead (NM) [50] and Adaptive Nelder–Mead (ANM) [51], were also included in the comparison because in earlier studies they were used to adjust the parameters of Hill’s muscle model [11,23,25]. To measure performance, 10 sets of perturbed muscle parameters were generated as the initial model parameters using uniformly distributed additive noise with a lower bound of −5 [cm] and an upper bound of 5 [cm]. 

These noise terms were added to the value of the identified parameters listed in Table 3. In the experiment, each method shared the same 10 sets of the perturbed muscle parameters (24 muscle parameters in a set, thus, 240 in total), and the parameters were optimized for the same subject’s measured trajectory. Figure 6 shows the predicted trajectory and torque identified from the 10 sets of the perturbed muscle parameters. 

The solid lines represent the average position and torque determined from the 10 sets of the identified muscle parameters, and the shadow area maps the maximum and minimum values for the angle and torque. The dotted lines represent the subject’s measured position and torque, used as the convergence target for optimizing the muscle parameters. As seen in Figure 6A, each method’s predicted trajectory (solid lines) was close to the reference position (dashed line). 

The same is seen for the torque (Figure 6B). Thus, each method could optimize muscle parameters that fit the subject’s movement trajectory comparably. Furthermore, the shaded areas of each method were small, meaning that the 10 sets of the muscle parameters identified by each method generated trajectories and torques that converged to the reference trajectory and torque.

Although the movement trajectories and torques predicted by each method were convergent, the identified muscle parameters may be divergent. Since the movement trajectory was from the same subject, even different initial muscle parameters should converge to similar muscle parameters. We compare the convergence results of the analytical gradient method (Equation (23)) and the gradient-free methods for the optimal muscle length in VM (Figure 7A) and ST (Figure 7B). During the iterative process, the shaded area was gradually reduced by the analytical gradient. In contrast, the shaded area was only slightly reduced and could even be diverged by the two gradient-free method. A similar observation was made for the calculated tendon slack lengths for VM and ST as shown in Figure 7C,D, respectively.

Predicting the trajectory and torque using Hill’s muscle model was complex because it involved numerical integration. Prior studies usually adopted gradient-free methods to update muscle parameters, thereby, avoiding the complexity of the analysis and the computational effort associated with numerical integration [23,25,31]. To compare the execution time of the analytical gradient method with alternative methods, we selected two gradient-free methods, NM and ANM, and three gradient approximation methods, CG [52], Truncated Newton (TNC) [52,53], and Sequential Least-Squares Programming (SLSQP) [54]. 

In the CG, TNC, and SLSQP, the two-point finite-difference method (FDM) was used to approximate the gradient [55], the popular method for the numerical approximation of the gradient. The results of this comparison are summarized in Table 4. Not surprisingly, the two gradient-free methods (NM and ANM) took the shortest average execution time and lowest number of evaluations. While the three gradient approximation methods (SLSQP, TNC, and CG) performed much worse, our analytical gradient method was competitive with the gradient-free methods. Interestingly, although the CG evaluated the loss most frequently on average, its execution time was sublinear in iterations because the just-in-time (JIT) compilation’s cache function compressed it.

## 4. Discussion

In this study, we proposed a differentiable continuous system control that integrates the dynamics of musculoskeletal models and analyzed the interaction mechanisms between the subject and the exoskeleton to achieve assist-as-needed control. This approach has several advantages, including faster convergence, increased accuracy and stability of predicted outcomes. For convergence, both muscle contraction and rigid body dynamic equations were applied in our model to improve calculating gradients. Updating the direction based on the loss value around the parameter made convergence faster in our model than in gradient-free methods, such as the NM method [50].

We considered both the muscle contraction and the rigid body dynamic equation to calculate analytical gradients in our model, and the calculated values of the muscle parameters (Table 3) were close to those reported by others [22,46,56]. The optimization results (Figure 4) showed that the trajectory and torque obtained from the adjusted muscle parameters were close predictions of the measured positions and torques. In earlier models [11,36,37], the torque was calculated from the motion equation through inverse dynamics transformation, which takes longer to compute. 

In comparison, the step of inverse dynamics transformation is eliminated in our model, and calculating the torque only requires the measured angle for parameter estimation. According to the results in Figure 6, the torque derived from the predicted trajectory through the inverse equation of motion afterwards was similar to the torque expected from the reference trajectory. This means that our training framework can implicitly and automatically estimate the torque during motion.

The iterative update of the muscle parameters was performed using the analytic gradient based on the muscle contraction and rigid body dynamic equations. Unlike the gradient-free methods, the analytical gradients method makes use of the dynamic equation and can generate the specified torque and angle leading to stable convergence of the muscle parameters as shown in Figure 7. 

The analytical gradient obtained the convergent results by calculating the gradients from the muscle contraction and motion equation with the two dynamic equations putting constrains on the results. In contrast, NM and ANM were gradient-free methods, relying only on the magnitude of the changes in the loss value to determine the direction of parameter update. The benefit of using the muscle contraction and motion equation is that the outputs satisfy the laws of physics. Thus, the use of analytical gradients to estimate muscle parameters could provide a more fault-tolerant and robust initial parameter setting process in practice. 

We compared the execution time for different parameter-updating methods (Table 4). The gradient approximation methods required multiple evaluations per iteration. This suggests that the simple two-point finite-difference-method (FDM), which was used to compute each parameter separately, becomes impractically time-consuming when applied to calculating loss functions in a high-dimensional parameter space. In contrast, the analytical gradient has an analytic solution, and for each iteration, it requires only two evaluations of the loss function: forward inference and backward propagation. The bottleneck in the computation time was in the evaluation of the loss function (Equation (17) in the Section 2.5), which involved time-consuming numerical integrals. 

Therefore, to compare the speed of the different update methods, we selected to evaluate only the loss function (the same for all methods) on the same hardware platform (an Apple Macbook pro M1 CPU; eight cores). The programs were implemented with the JAX package [56] and accelerated with JIT compilation. The results of this experiment (Table 4) showed that the analytical gradients method was competitive with the gradient-free methods in execution time.

Our proposed model merged a differentiable continuous system with a dynamic musculoskeletal model to support an effective and accurate performance for an exoskeleton controlled by EMG signals. This approach could be used to shorten the training time for controlling exoskeleton across different subjects. However, there are limitations to our study. First, this was a pilot study that included a small number of subjects. Second, the EMG signals used to control the model were recorded from only five healthy subjects. More subjects, including healthy and unhealthy ones, will have to be included in a follow-up to our study.

## 5. Conclusions

To overcome the time lag between the user intention and exoskeleton motion, we applied a differentiable continuous systems approach to a dynamic musculoskeletal model using EMG signals. The trajectories, including the torque and angles during muscle contraction, predicted by the proposed model were similar to the real movement. Moreover, our model outperformed others regarding the convergence rate and execution time. Our study demonstrates an effective and accurate method to decode EMG signals that serves the exoskeleton application. Future extensions of our proposed framework can be for deep neural networks with backpropagation and providing forward inference with biomechanical explanations.

## Figures and Tables

**Figure 1 biosensors-12-00312-f001:**
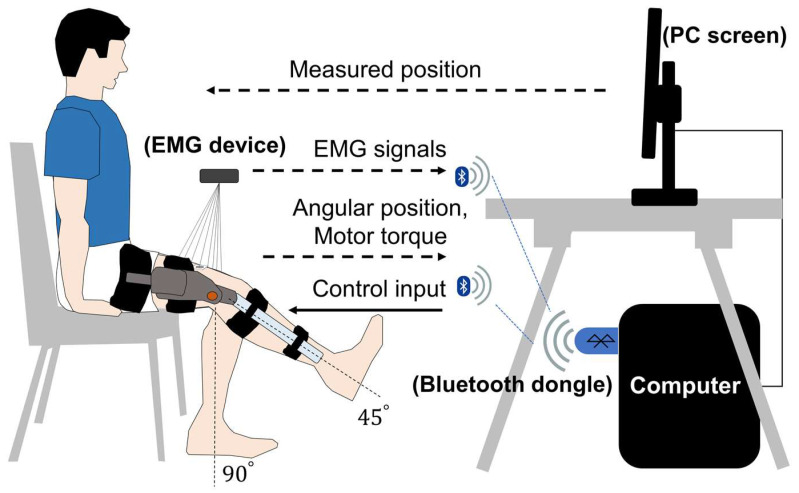
Illustration of the human-in-the-loop environment for the wearable knee exoskeleton. The position, θ(t), of the observed subject was measured to adjust the subject’s lower-limb force. A control algorithm adjusted the control input based on the EMG signals, e(t), position, θ(t), and motor torque, $\tau_{e}\left(t\right)$. The signals generated in this environment were transmitted by using the Bluetooth protocol.

**Figure 2 biosensors-12-00312-f002:**
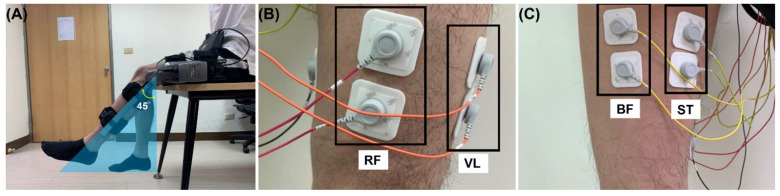
The knee exoskeleton and the electrode positions targeting specific muscles. (**A**) The participant performs the rotatory movement at the knee with a flexion–extension angle of 45 degrees while wearing the knee exoskeleton. (**B**) Positions of the surface electrodes for RF and VM (left thigh, anterior view). (**C**) Positions of the surface electrodes for BF and ST (left thigh, posterior view).

**Figure 3 biosensors-12-00312-f003:**
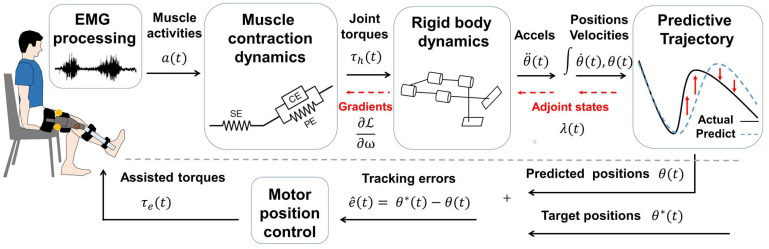
Schematics of differentiable musculoskeletal parameter estimation and control in the closed-loop system. Above the gray dashed line indicates the stage of the parameter estimation, which was performed offline by using the recorded EMG signals and trajectory positions. The muscle parameters were updated by using the gradient derived from the analysis, depicted by the black solid line and the red dashed line. Below the gray dashed line indicate the control stage, represented by a solid black line, use the identified musculoskeletal parameters to predict the trajectory positions and adjust the assisted torques.

**Figure 4 biosensors-12-00312-f004:**
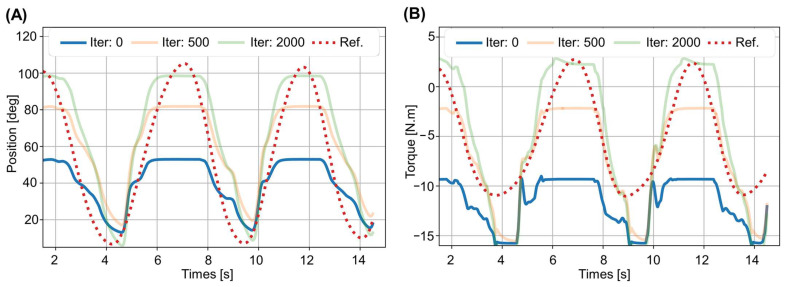
The process of optimizing muscle parameters by the iterative improvement of the predicted trajectory and torque. (**A**) Predicted position plotted against the iteration number. (**B**) Predicted torque plotted against the iteration number.

**Figure 5 biosensors-12-00312-f005:**
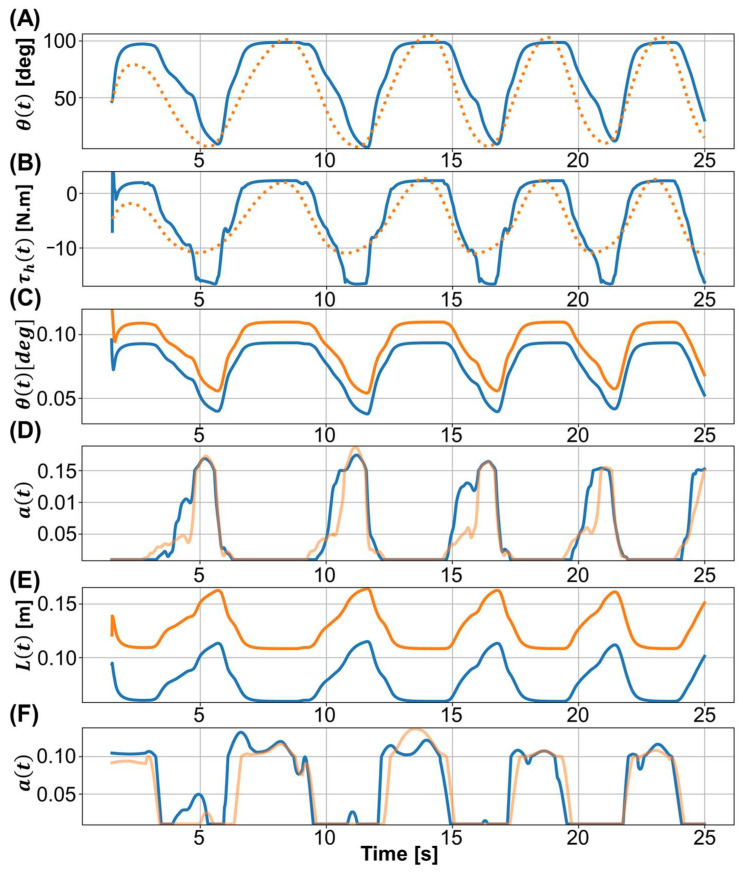
(**A**) The predicted trajectory (solid line) and measured trajectory (dashed line). (**B**) Total torque of RF, VM, ST, and BF. The predicted torque (solid line) and the reference torque (dashed line) were calculated from (**A**) using the Equation (6). (**C**) Predicted muscle length of RF (blue) and VM (orange). (**D**) Muscle activation of RF (blue) and VM (orange) converted from EMG signals. (**E**) Predicted muscle length of ST (blue) and BF (orange). (**F**) Muscle activation of ST (blue) and BF (orange) computed from EMG signals (Equation (2)).

**Figure 6 biosensors-12-00312-f006:**
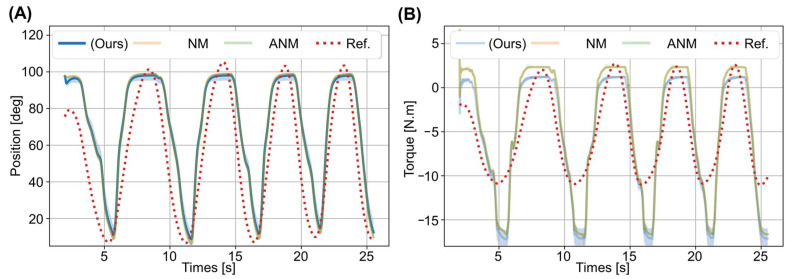
Convergence analysis of predicted trajectory and torque. (**A**) Predicted position. (**B**) Predicted joint torque.

**Figure 7 biosensors-12-00312-f007:**
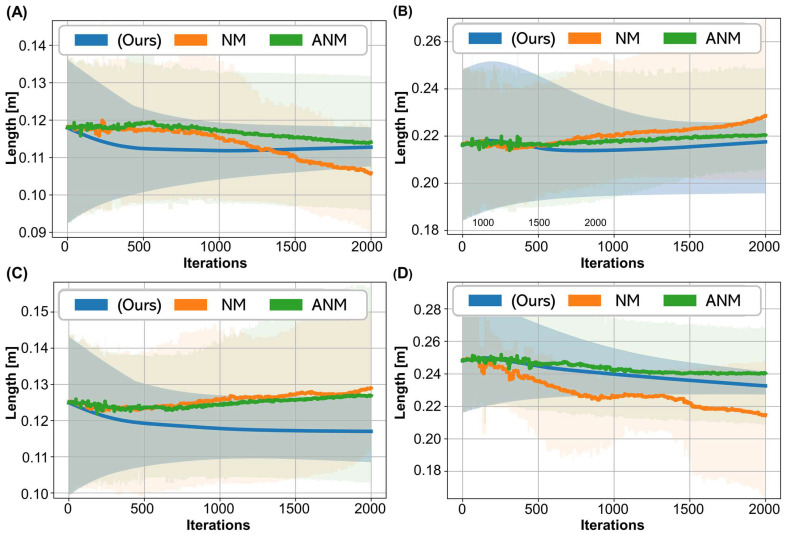
Convergence analysis of the identified muscle parameters. (**A**) Optimal muscle fiber length in VM. (**B**) Optimal muscle fiber length in ST. (**C**) Tendon slack length of VM. (**D**) Tendon slack length of ST.

**Table 1 biosensors-12-00312-t001:** List of the time-dependent parameters of the adopted muscle contraction and rigid-body dynamics.

Symbol	Description	Equation Number
e(t−d)	Filtered EMG signal *d* time steps earlier	(1)
** u(t−1) **	Neural activation one time step earlier	(1)
** $a^{\left(j\right)}\left(t\right)$ **	Muscle activation of *j*-th muscle at time *t*	(6)
** $l^{\left(j\right)}\left(t\right)$ **	Muscle length of *j*-th muscle at time *t*	(6)
$F_{mt}^{\left(j\right)}\left(a^{\left(j\right)}\left(t\right),l^{\left(j\right)}\left(t\right)\right)$	Muscle–tendon force of *j*-th muscle at time *t*	(6)
** $\tau_{g}\left(t_{i}\right)$ **	Gravitational torque at the *i*-th sample of time *t*	(3), (4)
** $\tau_{e}\left(t_{i}\right)$ **	Exoskeleton torque at the *i*-th sample of time *t*	(3), (4)
** $\tau_{h}\left(t_{i}\right)$ **	Human (or muscle) torque at the *i*-th sample of time *t*	(3), (4), (20)
$\theta \left(t_{i}\right)$ $,\;\dot{\theta }\left(t_{i}\right),\;\ddot{\theta }\left(t_{i}\right)$	Angular position, velocity, and acceleration, respectively, at the *i*-th sample of time *t*	(3), (4), (20)
** $h\left(\tau_{h}\left(t_{i}\right)\right)$ **	Predicted angular acceleration at the *i*-th sample of time *t*	(20)
** $\dot{z}\left(t_{i},\tau_{h}\right)$ **	Predicted angular velocity at the *i*-th sample of time *t*	(20)

**Table 2 biosensors-12-00312-t002:** The five identified parameters of the wearable exoskeleton in the motion equation.

$J\;\left(kg.m^{2}\right)$	$B\;\left(\frac{Nm\cdot s}{rad}\right)$	$K\;\left(\frac{Nm}{rad}\right)$	m (kg)	$r_{cm}\;\left(m\right)$
** 0.07 **	1.5	1.27	5.5	0.16

**Table 3 biosensors-12-00312-t003:** The identified muscle parameters in Hill’s model.

Groups	Muscles	Comparison	$L^{⋆}\;\left(cm\right)$	$l_{slack}\;\left(cm\right)$	$F_{\max}\;\left(N\right)$
RF	Rectous femoris	_	9.8	32.8	850
		Prev	7.6	34.6	848
VM	Vastus lateralis	_	9.6	12.9	2260
		Prev	9.9	13.0	2255
	Vastus medialis	_	10.6	12.4	1445
		Prev	9.7	11.2	1443
	Vastus intermedius	_	11.5	11.3	1025
		Prev	9.9	10.6	1024
ST	Semimembranosus	_	13.4	42.3	1092
		Prev	8.0	34.8	1162
	Semitendinosus	_	22.3	22.7	315
		Prev	20.1	24.5	301
BF	Biceps long head	_	8.4	31.9	701
		Prev	9.8	32.2	705
	Biceps short head	_	9.5	9.1	327
		Prev	11.0	10.4	315

In the Comparison column, a dash (_) indicates the row of parameters identified in our study and the abbreviation Prev indicates the parameters measured in a previous study [46].

**Table 4 biosensors-12-00312-t004:** Comparison of the computational efficiency and speed of gradient-free, gradient approximation, and gradient analytic methods.

	NM	ANM	SLSQP	TNC	CG	(Ours)
Gradient	free	free	approx.	approx.	approx.	analytic
#Eval/iter	1.3	1.4	29.4	125	533	2
Sec/iter	0.08	0.09	0.56	2.51	4.72	0.15

The number of evaluations (#Eval/iter) and execution time in seconds (Sec/iter) of the loss function in each iteration. Each value in the table was calculated by averaging 100 iterations.

## Data Availability

All recoded EMG data in this study are made available at the online repository (https://doi.org/10.5281/zenodo.6516777), assessed on 29 April 2022.

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
