# Peer review of "A Differentiable Dynamic Model for Musculoskeletal Simulation and Exoskeleton Control"

_biosensors, 2022, doi:10.3390/bios12050312_

Round 1

Reviewer 1 Report

Please carefully revise it. I attached the report of the comments. 

Reviewer 2 Report

Well the paper is very interesting but I would like to address the attention of the authors to three possible (I feel necessary) improvements.

a) first is the mathematical description of the model: presentation of equations is difficult to follow also because the various functions are not well indicated along with subscripts and superscripts; this part of the paper (paragraph 2.2) need a substantial rewriting with clarification

b) I would suggest to present a description of the system and feedback using a more formal control theory format: using Laplace transform would be better, although time domain description might be used for system parameters' identification

c) I wonder why the authors failed to cite the old and seminal papers of Duane T. McRuer which might be simply find here: https://scholar.google.com/scholar?hl=en&as_sdt=0%2C5&q=d+t+mcruer&btnG=

Regarding to point c) above, the authors will find that their work is rooted in McRuer's works and the new work might take a lot of benefit from those old results.

English text should be revised.

Round 2

Reviewer 1 Report

Dear authors, I appreciate your efforts. But you didn’t add the Q-5 in your section. I said that please add one new section in the name of Related work and write the literature separately. I also suggested article related to EMG, I didn’t found his work in the paper. 
      Kindly carefully check and Add the new section (2. Related Work) and follow the Q5 comment. 

Reviewer 2 Report

I think the paper has been substantially ameliorated and very good for publication.

Author Response

We thank for the reviewer's kindly comments.